# Effect of nutrition survey 'cleaning criteria' on estimates of malnutrition prevalence and disease burden: secondary data analysis

Sonya Crowe[1], Andrew Seal[2], Carlos Grijalva-Eternod[2] and Marko Kerac[2,3]

[1] UCL Clinical Operational Research Unit, Department of Mathematics, London, UK
[2] UCL Institute for Global Health, Institute of Child Health, London, UK
[3] Leonard Cheshire Disability and Inclusive Development Centre, Department of Epidemiology and Public Health, University College London, UK

Corresponding author
Andrew Seal, a.seal@ucl.ac.uk

## ABSTRACT

Tackling childhood malnutrition is a global health priority. A key indicator is the estimated prevalence of malnutrition, measured by nutrition surveys. Most aspects of survey design are standardised, but data 'cleaning criteria' are not. These aim to exclude extreme values which may represent measurement or data-entry errors. The effect of different cleaning criteria on malnutrition prevalence estimates was unknown. We applied five commonly used data cleaning criteria (WHO 2006; EPI-Info; WHO 1995 fixed; WHO 1995 flexible; SMART) to 21 national Demographic and Health Survey datasets. These included a total of 163,228 children, aged 6–59 months. We focused on wasting (low weight-for-height), a key indicator for treatment programmes. Choice of cleaning criteria had a marked effect: SMART were least inclusive, resulting in the lowest reported malnutrition prevalence, while WHO 2006 were most inclusive, resulting in the highest. Across the 21 countries, the proportion of records excluded was 3 to 5 times greater when using SMART compared to WHO 2006 criteria, resulting in differences in the estimated prevalence of total wasting of between 0.5 and 3.8%, and differences in severe wasting of 0.4–3.9%. The magnitude of difference was associated with the standard deviation of the survey sample, a statistic that can reflect both population heterogeneity and data quality. Using these results to estimate case-loads for treatment programmes resulted in large differences for all countries. Wasting prevalence and caseload estimations are strongly influenced by choice of cleaning criterion. Because key policy and programming decisions depend on these statistics, variations in analytical practice could lead to inconsistent and potentially inappropriate implementation of malnutrition treatment programmes. We therefore call for mandatory reporting of cleaning criteria use so that results can be compared and interpreted appropriately. International consensus is urgently needed regarding choice of criteria to improve the comparability of nutrition survey data.

Subjects Epidemiology, Global Health, Nutrition
Keywords Nutrition survey, Data cleaning, Disease burden, Malnutrition prevalence

## INTRODUCTION

Child malnutrition is a major global public health problem (*FAO, WFP, IFAD, 2012*). Figures published in 2012 estimate that 314 million children younger than 5 years are stunted (chronically malnourished, with a low height-for-age) and 258 million are underweight (low weight-for-age) (*Stevens et al., 2012*). Acute malnutrition, which includes wasting (low weight-for-height or low mid upper arm circumference) and nutritional oedema affects much smaller numbers of children, but is of particular concern due to high case fatality (*Schofield & Ashworth, 1997*; *Heikens et al., 2008*). Severe wasting alone affects just 19 million children (*Black et al., 2008*) but causes over one million (*WHO/WFP/UNSCN/UNICEF, 2007*) of a total estimated 7.6 million under-five deaths per year globally (*Liu et al., 2012*). There are also adverse implications for survivors, as wasting in early childhood is associated with later stunting (*Richard et al., 2012*). This, in turn, has adverse cognitive and developmental implications, affecting individuals and ultimately, whole societies (*Grantham-McGregor, 2002*; *Grantham-McGregor et al., 2007*). The case to tackle malnutrition is clear. It is high on the international health agenda and is being championed by various international initiatives such as the Scaling up Nutrition (SUN) movement (*SUN, 2010*) and the global rollout of treatment programmes such as Community Management of Acute Malnutrition (CMAM) (*WHO/WFP/UNSCN/UNICEF, 2007*; *Collins, 2007*).

To be maximally effective, nutrition, health and other programmes with growth-related outcomes need to be informed by high quality evidence and supporting data. Malnutrition prevalence is a key statistic used by many players for many purposes including: informing and shaping international policy and politics; global and local needs assessment—deciding whether and where nutrition programmes need to be opened (and when they can close); monitoring and evaluation—determining whether particular interventions have been successful; and tracking progress towards major national and international targets such as the Millennium Development Goals (MDGs) (*Gillespie, 2003*; *UNHCR, 2008*).

Nutrition surveys, from which the all-important malnutrition prevalence statistics are derived, generally follow similar and standardised methodologies for the collection and reporting of data (*SMART, 2013*; *DHS, 2013*). However, one aspect of analysis and reporting stands out. Nutrition survey "data cleaning criteria" are rarely described in reports, yet are almost always applied to raw data. Their purpose is to exclude very high or low values which are more likely to represent measurement or data error than a truly very large or very small child. They are particularly useful when it is not possible to return to the field to review individual children directly. While there are many reasonable and justifiable ways to decide on the inclusion/exclusion of an individual child or individual measure of growth, Table 1, there is no one gold standard which is applied in all settings and all situations. Even in the same context, there is every chance that different analysts may select different criteria. The effect of these different choices on the direction and magnitude of malnutrition estimates is currently unknown.

In this paper, we therefore aimed to quantify how different, commonly used data cleaning criteria, affect nutrition survey wasting prevalence statistics.

**Table 1  Cleaning criteria: five methods currently in use for cleaning survey data prior to calculation of malnutrition prevalence.**

| Cleaning method | Statistical probability criteria *Exclude if:* | | | Biological plausibility criteria *Exclude if:* | Reference mean |
|---|---|---|---|---|---|
| WHO (2006) Growth standards (*WHO, 2006b*) | HAZ < −6 HAZ > 6 | WAZ < −6 WAZ > 5 | WHZ < −5 WHZ > 5 | - | Growth Standards |
| SMART flags[*] (*SMART, 2013*) | HAZ < −3 HAZ > 3 | WAZ < −3 WAZ > 3 | WHZ < −3 WHZ > 3 | - | Survey Sample |
| WHO 1995 Flexible criteria[**] (*WHO, 1995*) | HAZ < −4 HAZ > 3 | WAZ < −4 WAZ > 4 | WHZ < −4 WHZ > 4 | - | Survey Sample |
| WHO 1995 Fixed criteria (*WHO, 1995*) | HAZ < −5 HAZ > 3 | WAZ < −5 WAZ > 5 | WHZ < −4 WHZ > 5 | - - | Growth Reference |
| Epi-Info (*WHO, 2006a*) | HAZ < −6 HAZ > 6 | WAZ < −6 WAZ > 6 | WHZ < −4 WHZ > 6 | HAZ > 3.09 and WHZ < −3.09 HAZ < −3.09 and WHZ > 3.09 | Growth Reference |

**Notes.**

HAZ, Height-for-age z-score; WAZ, Weight-for-age z-score; WHZ, Weight-for-height z-score.

[*] The upper and lower values are flexible, i.e., can be increased based on judgment (*WHO, 2006b*).

[**] Recommended for use when the observed mean *z*-score is below 1.5 (*WHO, 1995*).

## MATERIALS & METHODS

### Study population and sample size

We performed secondary analysis of 21 national demographic and health survey (DHS) datasets, each with anthropometric data collected using standard DHS methods. We chose this dataset as they represent countries from the Lancet series with a high burden of disease. The dataset has a reference population of 36 countries, which account for the majority of the global malnutrition disease burden (*Black et al., 2008*). The 21 were those which had available nutrition surveys done in the last ten years. Each DHS survey size is large enough for robust national prevalence estimates (*Aliaga and Ren, 2006*). In total, the 21 DHS surveys comprised $n = 216,841$ children (after $n = 38,136$ records with missing age variables had been removed).

DHS survey methods are well standardised, both within-country and between countries (*DHS, 2014a*).

For surveys which we have analysed, data would have been recorded on paper in the field, then entered on a database which underwent a thorough process of checking and processing (*DHS, 2014b*). Data processing errors should, therefore, have been rare.

### Variables

Weight-for-age (WAZ), height-for-age (HAZ) and weight-for-height (WHZ) Z-scores based on WHO growth standards (*WHO, 2006b*) had previously been calculated (*Kerac et al., 2011*) from weight, height/length, age, and sex variables using Emergency Nutrition Assessment software; which was developed for the Standardised Monitoring and Assessment of Relief and Transitions (SMART) initiative (*ENA, 2012*). Any records with missing WAZ, HAZ, or WHZ were removed ($n = 13,545$). The mean WAZ, HAZ,

**Table 2  Case definitions of wasting.**

|  | Case definition |
|---|---|
| Wasting | WHZ[*] $< -2$ |
| Moderate wasting | WHZ $< -2$ and WHZ $\geqslant -3$ |
| Severe wasting | WHZ $< -3$ |

Notes.
[*] WHZ, Weight-for-height z-score, which represents standard deviations below the WHO growth standard mean (e.g., a Z-score of $-1$ = 1 standard deviation below the reference mean).

and WHZ for each country were calculated for children aged 6–59 months using the appropriate DHS sample weights.

## Cleaning criteria

Extreme anthropometric values are considered more likely to represent measurement or database errors than an individual who is truly very small or very large. The cut-offs for defining extreme values depend on the data cleaning method adopted, which may be based on the mean Z-score of either the reference population ('fixed criteria') or the observed data ('flexible criteria'). We compared five methods that are currently in widespread use (Table 1), three of which are 'fixed' criteria (*WHO, 2006a*; *WHO, 1995*; *Centres for Disease Control and Prevention, 2008*) and two 'flexible' (*SMART, 2013*; *WHO, 1995*). Note that the Z-score is defined by the standard deviation of the reference dataset rather than that of the sample in both the flexible and fixed methods. We note that the WHO recommend their flexible criteria is adopted when the observed mean Z-score is below $-1.5$, otherwise they recommend use of fixed criteria (*WHO, 1995*). While none of the 21 DHS surveys studied in this work had a mean WHZ less than $-1.5$, we applied both fixed and flexible criteria to allow for comparison. It should also be noted that criteria were applied in a manner that replicates their implementation in commonly used nutritional survey software when calculating the prevalence of wasting. Therefore, when using WHO and SMART criteria, outliers were only excluded using WHZ thresholds. However, when using EpiInfo criteria, WHZ, WAZ, and HAZ thresholds were used to identify outliers. In addition, when using EpiInfo criteria exclusions were made on the basis of biological implausibility criteria, i.e., incompatible combinations of HAZ and WHZ, HAZ $> 3.09$ and WHZ $< -3.09$, or HAZ $< -3.09$ and WHZ $> 3.09$ (*Dean et al., 1991*).

## Data handling and analysis

Analysis was performed using Stata version 12 (StataCorp., TX), using the appropriate sample weights defined by DHS. Wasting and severe wasting prevalence based on current case definitions (Table 2) were estimated for each country, excluding records according to each cleaning criteria in turn. The standard deviation of weight-for-height measurements, after the data was cleaned, was then calculated for each country.

Country-level wasting prevalence was compared to the international 'integrated food security phase classification' (IPC) (*IPC Global Partners, 2008*), which is used to determine the severity of an emergency and guide the need for interventions (Table 3). We emphasise

**Table 3 International 'integrated food security phase classification' (IPC) version 2.**

| IPC classification of food insecurity level | Prevalence of wasting (Weight-for-height z-score < −2) |
|---|---|
| Minimal | < 5% |
| Stressed | 5–10% |
| Crisis | 10–15% |
| Emergency | 15–30% |
| Famine | >30% |

that the IPC anthropometric thresholds are not normally used on their own to classify emergency situations. They serve here to demonstrate the extent of differences between cleaning criteria.

To illustrate the implications for treatment programmes, we estimated the caseload over a 1 year period for severe wasting (WHZ < −3) based on the methodology proposed by *Myatt (2012)* for each of the cleaning criteria in turn, using the formula:

Caseload for severe acute malnutrition (SAM) $= N \times P \times K \times C$,

where $N$ is the size of the population in the program area; $P$ is the estimated prevalence of SAM; $K$ is a correction factor to account for new cases over the 1-year period; $C$ is the expected mean program coverage over the 1-year period.

We focused here on severe wasting because it is the form of malnutrition with the highest case fatality—but with effective treatment programmes available (*WHO/WFP/UNSCN/UNICEF, 2007*). Since they are not reported in DHS surveys, we could not take into account low mid upper arm circumference (MUAC) or bilateral pitting oedema. These criteria, together with severe wasting, are independent case definitions for severe acute malnutrition (SAM). Population statistics were taken from the 2004 United Nations population database (*United Nations Statistics Division Demographic Yearbook, 2007*) and we assumed a mean programme coverage over the period of 50%.

### Ethics statement

This is a secondary analysis of freely available data so formal ethical clearance was not needed. Permission to use and analyse the dataset was obtained by registering the project on the Demographic and Health Survey (DHS) website via http://www.measuredhs.com/accesssurveys/access_instructions.cfm.

## RESULTS AND DISCUSSION

### Databases characteristics

In the 21 surveys, we had a total sample size of 163,228 children aged 6 to 59 months. These were representative of a total estimated population of 211 million children. Samples varied relative to country size: India's DHS survey had the largest sample (45,398 children) and Cote D'Ivoire the smallest (1,710 children). Full details of the DHS surveys used in the

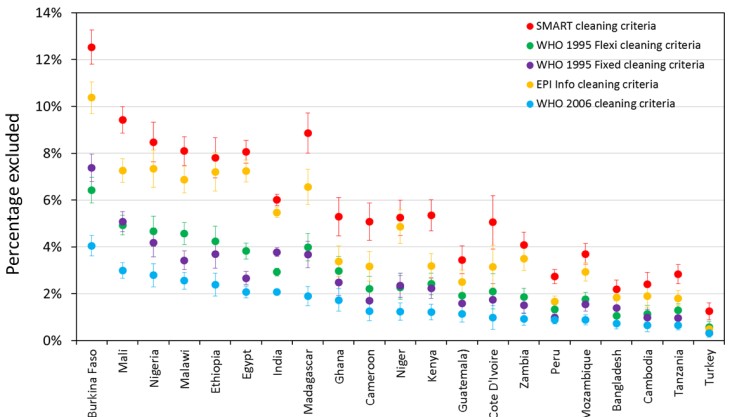

**Figure 1** Percentage of records excluded from prevalence estimates for children aged 6–59 months on the basis of five different cleaning criteria, by country.

analysis are presented in a web appendix (Table S1), including: country, year of survey, sample size for children aged 6–59 months, and the estimated population of children aged 6–59 months.

## Impact of different cleaning criteria on record exclusion

Figure 1 shows the percentage of records excluded from prevalence estimates on the basis of five different cleaning criteria, by country (for children aged 6–59 months). SMART criteria consistently exclude the most children and WHO 2006 criteria exclude the least. However, what difference this makes, in terms of the absolute and relative proportion of exclusions, varies markedly by country. In Burkina Faso 4.1% were excluded by WHO 2006 and 12.5% were excluded by SMART, while in Turkey 0.3% were excluded by (WHO-2006) and 1.3% by SMART.

## Impact of different cleaning criteria on reported prevalence

Prevalence estimates for wasting and severe wasting of children 6–59 months are shown under different cleaning criteria, by country, in Fig. 2 and Fig. 3 respectively. The coloured boundaries in Fig. 2 relate to the international IPC classification. Again, there is marked variation in different countries. In many countries, whilst the absolute prevalence figure varies according to cleaning criterion, the IPC category does not change. In other countries, however, the application of different cleaning criteria results in the crossing of a phase boundary and a different categorization of 'severity'. The proportional differences in severe wasting (Fig. 3) are greater than for total wasting (Fig. 2).

## Implications for clinical caseload planning

The implications of the SAM prevalence differences are shown in detail in web appendix (Table S2). Estimated clinical caseloads for SAM for every country under each cleaning criteria can be seen to vary greatly. For example, were no cleaning criteria applied for India, the country would have to plan for some 10.4 million SAM cases; with WHO-2006 criteria, for 8.7 million cases; and with SMART flags, for only 6.5 million cases. Large differences
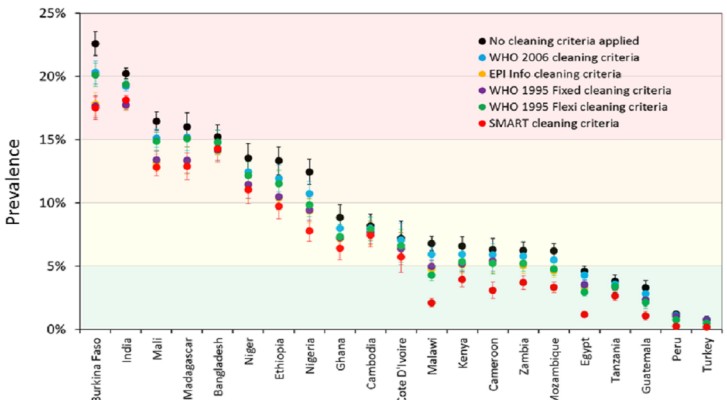

**Figure 2  Prevalence of wasting (WHZ < −2) for children 6–59 months under different cleaning criteria, by country.** The coloured boundaries relate to the international 'integrated food security phase classification' (IPC) (see Table 3).

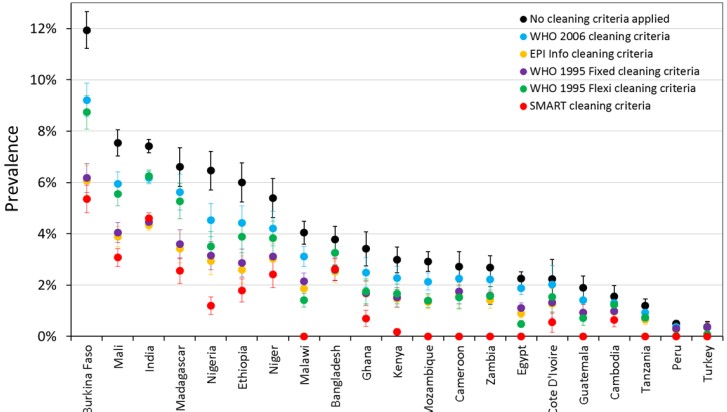

**Figure 3  Prevalence of severe wasting (WHZ < −3) for children 6–59 months under different cleaning criteria, by country.**

in estimated caseload are observed in many counties when the least inclusive cleaning criteria are compared with the most inclusive cleaning criteria. In 9 countries, SMART flags exclude all potential cases of SAM.

## Explaining the observed differences

Figure 4 shows prevalence estimates under the (least inclusive) SMART criteria plotted against the standard deviation of the WHZ distribution before data exclusion. Each point represents one of the DHS surveys (not labelled), and is either wasting prevalence (black) or severe wasting prevalence (blue). The observed linear relationship helps to explain why we see the differences we do: the wider the survey distribution (due to either lower survey data quality or the heterogeneity of the population), the more likely the extremes are to be excluded, hence the greater the difference in prevalence estimate. It is also important to note here that there may be children with incorrect anthropometric measurements that are within the plausible range and so are not removed by the cleaning criteria.

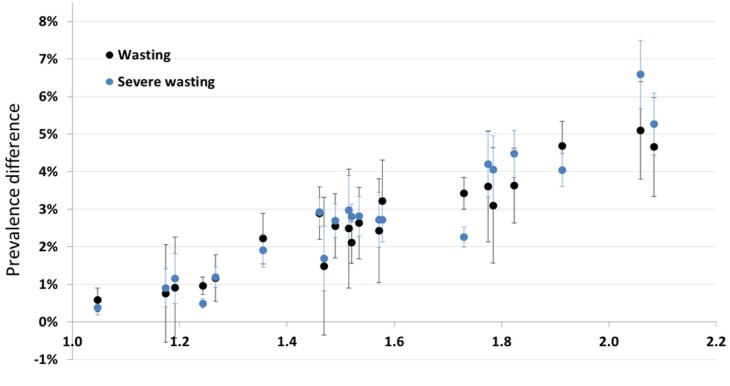

**Figure 4 Scatterplot of the difference between prevalence with no cleaning and SMART cleaning, versus the standard deviation of the WHZ distribution for non-cleaned data.** Each point is a country (not labelled): black points denote wasting (WHZ < −2) whilst blue points denote severe wasting (WHZ < −3).

## DISCUSSION

Our results show that the application of different cleaning criteria has a profound effect on the reported prevalence of both moderate and severe wasting. The magnitude of effect varies markedly between different countries, and is most pronounced for severe wasting. This in turn has marked effect on estimated programme caseloads. Since wasting prevalence is a key statistic but choice of cleaning criteria is not currently standardised, differences in practice between individual analysts could unduly influence the results that are made available to decision-makers. This may potentially lead to inconsistent, inefficient and inappropriate implementation of malnutrition treatment programmes.

This is, to our knowledge, the first paper to highlight this important issue. However, our paper has limitations and we call for others to follow-up and address these weaknesses and data gaps.

First, our results are based on a relatively limited number of countries. Since our DHS surveys represent countries which account for the highest burden of global malnutrition, and since there is no good reason to believe that they are not representative of others, we feel it unlikely that the overall message of our paper would change were more surveys included. However, it is important that the analysis is extended elsewhere, particularly to different types of datasets. DHS follow a standardised methodology and are mostly done in relatively stable environments. Surveys conducted in emergency settings, working under greater pressure, and sometimes by more inexperienced teams, may have more issues with data quality, so cleaning criteria may have even more effect on final results—but this would need to be confirmed.

Second, in our caseload calculations for therapeutic feeding, we have not been able to take into account the prevalence of oedematous malnutrition, nor that defined by low MUAC. These independent criteria for SAM also account for programme admissions and are not influenced by choice of cleaning criteria. So where, for example, the prevalence of oedematous malnutrition is high, the weight-for-height caseload estimates only form a

small part of the estimate of global acute malnutrition (GAM); even if wasting caseload doubles, it may not make a large overall contribution to caseload planning.

Another limitation is that we have focused here on the impact of anthropometric cleaning criteria on wasting, and have ignored stunting and underweight. Future work, however, should explore the effect of data cleaning on other forms of anthropometrically defined malnutrition. There is, for example, increasing recognition of the 'double burden of disease', with over-nutrition and under-nutrition co-existing in the same setting (*Grijalva-Eternod et al., 2012*). In some situations the application of cleaning criteria may well impact heavily on estimates of child overweight/obesity. Effects on stunting (low height-for-age) prevalence are also important to explore, it being a focus of much current international policy interest (*Stevens et al., 2012*; *SUN, 2010*). It is critical that any trends or inter-country differences are 'real' rather than artefacts of survey cleaning criteria choice. As well as inadvertent differences due to poor awareness of the effect of cleaning criteria, there is clearly also potential for deliberate "gaming", by which inclusive or exclusive criteria are deliberately chosen to fit political agendas (e.g., choose an inclusive criterion if you want the prevalence to appear high so as to justify external investment into your programme; choose an exclusive criterion if you want prevalence to appear low, so as to demonstrate the programme has 'worked'). One area for further work would be to examine whether the application of different cleaning criteria may have led to actual differences in food assistance responses.

Third, it is important to extend the analysis to other age groups. We have focused here on children aged 6–59 months since they are the main target group for therapeutic and supplementary feeding programmes. However, malnutrition can also be prevalent in older children (and even in adults in extreme situations). Infants aged <6 months are another group in whom the problem of malnutrition is increasingly being noted (*Kerac et al., 2011*). Cleaning criteria may have different effects on prevalence estimates in these other groups.

To address the problems we have raised, we propose several solutions. The first is a call for mandatory reporting of which cleaning criteria were used so that results may be interpreted accordingly. Our paper gives clear messages on which criteria are more 'inclusive' (and thus tend to give higher prevalence results) and which are more 'exclusive' (and thus give lower prevalence results). Any inter-survey or time trend differences can thus be accounted for as potentially due to/not due to (if the same criteria are used consistently) data cleaning practices.

Building on our work, it may be possible to establish equations by which prevalence calculated using one cleaning method could be "transformed" to an estimate using another method. This would, however, be at best an approximation and would only be needed if raw datasets were unavailable for full re-analysis (the current momentum to open-source datasets as well as results would be helpful here).

A third call is for urgent international consensus and guidance on selecting and adopting a single set of optimal cleaning criteria. This would improve the comparability of nutrition survey data (including trend data in the same setting) and the coherence of associated policy recommendations.

Finally, and for the longer term, we note the current trend to electronic data collection. This may be particularly useful for nutrition surveys as extreme data values could potentially be validated in the field, reducing or eliminating the need for data exclusion during analysis.

We end by repeating our call for greater awareness of cleaning criteria as an explanation of inter-survey differences in malnutrition prevalence. This can affect policy decisions and planning and is therefore not just of importance to survey analysts, but is a much wider issue for policy makers and programme managers at local, national, and international level. With the emergence of the "double burden" of malnutrition, where we are seeing extremes of over-nutrition as well as under-nutrition, it is critical to address this problem as analysis and interpretation will become more complex and difficult over time unless dealt with now. For nutrition to continue to have the credibility it deserves as a public health priority, its statistics must be reliable, consistent, and transparent.

**Abbreviations**

| | |
|---|---|
| **DHS** | Demographic and Health Surveys |
| **HAZ** | Height-for-age z-score |
| **MDG** | Millennium Development Goals |
| **MUAC** | Mid Upper Arm Circumference |
| **SUN** | Scaling Up Nutrition |
| **WAZ** | Weight-for-age z-score |
| **WHO** | World Health Organization |
| **WHZ** | Weight-for-height z-score |

## ACKNOWLEDGEMENTS

We thank the Clinical Operational Research Unit (CORU) at University College London (UCL) for their support which enabled this work.

### Funding

Funding was received from the UCL Grand Challenges for Global Health small grant scheme. The funders had no role in study design, data collection and analysis, decision to publish, or preparation of the manuscript.

### Grant Disclosures

The following grant information was disclosed by the authors:
UCL Grand Challenges for Global Health.

### Competing Interests

The authors declare they have no competing interests.

## Author Contributions

- Sonya Crowe performed the experiments, analyzed the data, contributed reagents/materials/analysis tools, wrote the paper, prepared figures and/or tables, reviewed drafts of the paper.
- Andrew Seal conceived and designed the experiments, wrote the paper, prepared figures and/or tables, reviewed drafts of the paper.
- Carlos Grijalva-Eternod conceived and designed the experiments, wrote the paper, reviewed drafts of the paper.
- Marko Kerac conceived and designed the experiments, contributed reagents/materials/analysis tools, wrote the paper, reviewed drafts of the paper.

## Human Ethics

The following information was supplied relating to ethical approvals (i.e., approving body and any reference numbers):

The paper conducted secondary data analysis on anonymised and publicly available data sets. No ethical approval was required.

## Supplemental Information

Supplemental information for this article can be found online at http://dx.doi.org/10.7717/peerj.380.

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
