# Peer review of "Effect of nutrition survey ‘cleaning criteria’ on estimates of malnutrition prevalence and disease burden: secondary data analysis"

_PeerJ, doi:10.7717/peerj.380_

## Round 0.1 · original submission · Minor Revisions

The reviewers returned favorable comments. However, they requested some modifications which need attention. Please make the requested changes if you agree with them and return your manuscript along with responses to the reviewers.

Reviewer 1 ·

Basic reporting

No comments

Experimental design

No comments

Validity of the findings

No comments

Additional comments

The manuscript submitted by Crowe et al., address an issue of utmost relevance for public health in developing countries. The manuscript is well written and the results properly analyzed. My comments are mainly to complete the information for the reader:
1) Is there any cleaning method that could be considered the best? According to the results, could it be discussed which one seems to be the best?
2) In order to identify the best cleaning method, a gold standard to identify SAM should be available. Is this the available? Or at least are there candidates? This should be discussed.
3) The authors used already available data to perform the secondary analysis. Certainly they didn’t participate in the collection of the results. However, it would be useful to know more about the process. For instance, were the data entered manually? If this was the case, this was done once or twice. If the second was the case, how was the disagreement between the 1st and the 2nd data entry handled?
4) The abbreviation SAM is introduced in line 106, but it appear earlier in the text (line 98). It should be introduced the first time it appears.

·

Basic reporting

The tables and figures are appropriately labeled but the scale and the color scheme, particularly in figure 2, obscure the finer points of the data but do allow the trend to be apparent.

Experimental design

The article is well laid out with a clearly stated aim that is addressed in the results section and the implications cogently explored in the discussion.

Validity of the findings

This is a well written article that raises the important implications of data cleansing and how the chosen criteria may affect the response to an important situation or crisis. The authors rightly call for disclosure in data cleansing methodology and propose the creation of a correction model so that data which are cleansed in different ways may be more readily compared.

Over-nutrition is a growing concern is poorer countries, particularly in Central America. The authors appropriately mention the potential impact that data cleansing may also have when addressing this problem and this is definitely an area that could be focused on in future projects.

There is a call for a data-cleaning consensus and the hope that data validation in the field may reduce errors and the need for extensive cleansing.

Several minor questions for the authors are below:

- In the absence of a consensus statement which method of data-cleaning would you use or recommend be used for now?

- In emergency situations would you recommend using the more inclusive WHO criteria, which may overestimate the severity of the situation, allowing for data collection errors in a hostile or difficult environment?

- Are you able to give any examples where discrepancies in reporting have led to an improper response to a food shortage crisis?

---

## Round 0.2 · accepted · Accept

Your article is accepted - congratulations